# Influence of Scanning Strategy Parameters on Residual Stress in the SLM Process According to the Bridge Curvature Method for AISI 316L Stainless Steel

**DOI:** 10.3390/ma13071659

**Published:** 2020-04-03

**Authors:** Jiri Hajnys, Marek Pagáč, Jakub Měsíček, Jana Petru, Mariusz Król

**Affiliations:** 1Department of Machining, Faculty of Mechanical Engineering, Technical University of Ostrava, Ostrava 708 00, Czech Republic; marek.pagac@vsb.cz (M.P.); jakub.mesicek@vsb.cz (J.M.); jana.petru@vsb.cz (J.P.); 2Department of Engineering Materials and Biomaterials, Faculty of Mechanical Engineering, Silesian University of Technology, Gliwice 44-100, Poland; mariusz.krol@polsl.pl

**Keywords:** scanning strategy, residual stress, SLM, AISI316L, BCM

## Abstract

The present paper deals with the investigation and comparison of the influence of scanning strategy on residual stress in the selective laser melting (SLM) process. For the purpose of the experiment, bridge geometry samples were printed by a 3D metal printer, which exhibited tension after cutting from the substrate, slightly bending the samples toward the laser melting direction. Samples were produced with the variation of process parameters and with a change in scanning strategy which plays a major role in stress generation. It was evaluated using the Bridge Curvature Method (BCM) and optical microscopy. At the end, a recommendation was made.

## 1. Introduction

Additive manufacturing (AM), namely selective laser melting (SLM) technology, allows us to create complex components that can be customised in different ways (topological optimisation, lightweight construction, lattice structures, etc.). Components by SLM are near full density and also have mechanical properties almost the same as bulk material [1]. SLM technology works on the basis of melting the individual layers to each other (layer-by-layer) directly from metallic powder, which creates thermal gradients that permeate the previously molten layer. The material expands and contracts, resulting in residual stress (RS) [2,3]. RS then leads to part distortion, delamination and cracks. Other defects may occur in the SLM process. The most widespread defects, besides RS, are the balling effect, warping and dross formation [4,5,6].

Process parameters and scanning strategies can be varied to reduce the impact of RS. Alimardani et al. [7] have reported in their study a reduction in RS while reducing laser power. On the contrary, an experiment conducted by Wu et al. [8] shows that increasing laser power and scanning speed increases the length of the melt pool, which reduces the residual stress. The same conclusion was achieved by Vasinota [9], who concluded that temperature gradients decrease by as much as 20% as the scanning speed decreases, but he also points out that decreasing the scanning speed and laser power causes a significant change in the size of the melt pool. The study [10] shows that the largest component of residual stress is generated parallel to the scan vector and increases with its length. In other words, the longer the scan path is, the greater the internal stress evolves. Therefore, it is advisable to choose a scanning strategy that does not use a long scanning path.

Preheating of the substrate before printing also has a significant effect on the RS formation [11]. Setting the correct temperature is manifested by a decrease in RS. Preheating plays a huge role, especially for titanium alloys; Ali. et al. [12] were able to effectively eliminate RS by preheating the substrate to 570 °C. However, from a certain point of temperature, preheating begins to stop metallurgical phenomena. In the absence of such phenomena, the preheating temperature must be set high enough because the temperatures during the process are above 100 °C because of the thermal input of the laser itself.

Various methods have been developed for the measurement of RS, which differ in accuracy, in the size of required samples, and in the method of measurement (destructive or non-destructive). The non-destructive method of measuring RS is by means of a diffractometer (XRD—X-ray diffraction), using a crystalline lattice as a strain gauge and measuring the distance between crystallographic lattices of planes (d-distance), which depends on the deformation. Thus, the stresses can be determined from the measured d-distances [13]. However, the most widespread RS measurement technique is called the crack compliance method (CCM) and was developed by Michael Prime [14]. It is a technique based on the measurement of partial deformation after stress relief, where a strain gauge is attached to the sample. A similar technique is called the hole drilling method, when strain gauge induced stress is measured after a hole is drilled. A special method called the bridge curvature method (BCM) has been developed for AM [15]. This method is based on creating a bridge structure that is printed without supporting elements directly on the substrate, after which the structure is cut and the angle of deflection between the pillars is measured. This angle can be compared to an experimentally determined baseline or further processed by the simulation of the measured distortion in the finite element method (FEM) analysis. Because of the nature of the BCM method, RS can only be evaluated only in the X direction.

This study deals with the effect of setting print parameters and scan strategy parameters, where RS is evaluated using a modified BCM method. The bridge-shaped samples were made of AISI316L stainless steel, then evaluated using an optical measuring system. The aim was to find out the ideal setting of parameters of individual strategies and printing parameters depending on RS. This will make it possible to effectively reduce RS during the printing process itself.

## 2. Materials and Methods

### 2.1. Powder Characterisation

For experimental purposes, AISI316L powder supplied by Renishaw was used, which was produced by gas atomisation. It is a non-magnetic austenitic stainless steel, which contains a very low percentage of carbon and is alloyed with chromium, nickel, molybdenum and other negligible elements, the entire chemical composition specification of virgin powder is shown in Table 1.

Before the experiment itself, an analysis of the powder was carried out in terms of morphology, particle distribution and internal porosity. The morphology of the powder is an important characteristic that affects the recoater deposition of metal powder in terms of flowability and packing density. Morphology was performed using a scanning electron microscopy (SEM) on a JSM-6510 device (JEOL, Akishima, Japan). When evaluating the morphology of virgin powder, it was revealed that not all particles are perfectly spherical and that some particles contain satellites that are marked with red arrows (see Figure 1).

The term ‘porosity of powder’ refers to the internal porosity of the powder, but also refers to the surface porosity. However, in both cases, this is an undesirable behaviour that affects the degree of total porosity of the formed parts. This is due to the manufacturing nature of the gas atomisation method, where gas is trapped inside the particle. Internal porosity can be measured by optical microscopy on polished resin-embedded specimens and the powder particles which were cut in half [16]. There is also a non-destructive computed tomography method that scans a specimen in 3D and detects pores using X-rays. In this study, a destructive optical method was used, in which the powder sample was poured into cold resin; after curing, the 1 mm layer was abraded, thereby cutting the powder particles in half and exposing the internal porosity, then the specimen was polished with diamond paste. So far, there is no standard to measure the internal porosity of metallic powder that quantifies the measurement procedure, so the study used the KEYENCE microscope (VHX-5000, KEYENCE, Itasca, IL, USA) and the analysis software delivered with a microscope. The assumption is that, in general, pores must be closed from 3/4 of their circumference so that they can be considered pores [17].

### 2.2. Settings and Sample Fabrication

For the production of samples, a Renishaw AM400 3D printer (Wotton-under-Edge, Great Britain) was used, which has a laser on the maximum nominal power of 400 W. In the production of samples, the focus size was set to 70 µm, and argon with a purity of 5.0 was chosen as an inert gas. Because of the inert gas, air was pushed out from the chamber and kept below 1000 ppm throughout the printing process, ensuring that the powder would not oxidise during building and also due to the gas flow to properly remove metal vapours from the process of printing. The build was prepared using QuantAM software (5.0.0.135, Renishaw, Wotton-under-Edge, Great Britain). All samples were printed without support elements directly onto the substrate. A total of 48 samples were divided into two builds to make samples with substrate preheating and without preheating. The bridge samples started printing from the pylons and continued to the upper part.

The dimensions and geometry of the samples were inspired by the BCM shape [15], which was specially designed for the evaluation of RS metal parts made for AM. In this study, an improved sample shape was designed, see Figure 2a. For cutting, the samples from the substrate were the original shape designed to be used by WEDM, but the modified shape is more suitable for cutting by band saw. Improvements made to the design of Kruth et al. [15] consist of the creation of a special groove that acts as a guide groove for the band saw. This improvement contributes to faster cutting of samples and the groove also serves as a more accurate measurement and evaluation of the curling angle. Besides the faster cutting, it also had less local thermal influence on the sample than the WEDM method. The curling angle measurement, Figure 2b, and evaluation was performed on an Alicona Infinite Focus 5 optical instrument (Graz, Bruker Alicona, Austria). Layer thickness was 50 µm and rotation between layers was 67°. All other parameters remained at standard values.

### 2.3. Simulation of Chosen Samples

Before the experiment itself, several distortion simulations were performed in MSc Simufact 2020 (Hamburg, Germany), including simulation of cutting from the substrate. These simulations serve as a comparison and verification of accuracy using computing models. Software can also show deformation of the component and generate the Standard Triangle Language (STL) file, which has a preloaded deformation based on the simulation. To simplify deformation simulations, Simufact 2020 converts the model to a mesh model. Simulations allow setting the temperature of the preheating of the substrate, the laser power, the size of laser beam, scanning speed, layer thickness, and of course a large variety of materials, which is already preset in the library. To simulate individual scanning strategies, it is necessary to create a calibration cantilever that is cut in half right after printing to activate the RS, then these distortions are accurately measured and entered into the software. For this reason, the simulation was performed only on selected samples that were designated as reference samples. The simulation results are in a data file in STL format, which contains the deformation, see Figure 3a. The deformed part was measured and compared with the actual deformation measured after printing and the angle α was measured, see Figure 3b.

### 2.4. Design of Experiment

Design of experiment (DoE) according to the Taguchi method with the orthogonal array setting L16 (4 ^ 4) was used for sample production. The aim of the study is to investigate the effect of scanning strategies at various settings on the resulting RS. The Chessboard (CH), Meander (M) and Stripes (S) strategies were further tested, with variables of the laser power (P) and substrate preheating (T) also being tested in the experiment. The CH strategy varied the size of an individual islands, which helped to effectively reduce RS, but on the other hand, scanning by this strategy is very slow. By default, the island sizes are set to 5 mm^2^ and are rotated by 90 ° to each other, see Figure 4a. In the S strategy, the strip size varied to provide consistent heat distribution throughout the layer, see Figure 4b. For the last M strategy tested, it was not possible to vary the size of the island or the strip, so hatch distance (HD) was chosen as the tested parameter, see Figure 4c. For easier DoE evaluation, the hatch distance was designated as a field size (FS), but under other values (3—0.06 mm; 5—0.11 mm; 7—0.18 mm; 10—0.24 mm). This strategy is most widespread as it offers quick and efficient scanning and is mainly used for parts with a small XY cross section. Table 2 lists the parameters that have been varied according to Taguchi distribution. The same DoE design was used for each strategy separately, i.e., a total of 48 samples.

## 3. Results

### 3.1. Evaluation of Powder Characteristics

For the evaluation of internal particle density, KEYENCE software (5.2.0010, KEYENCE, Itasca, IL, USA) was used. From the analysis of several images taken by microscope, the total particle density of the powder was determined as 99.695%. Furthermore, the size of the individual internal pores was measured. The measurement was performed on images with a magnification of 1000× and again the software by KEYENCE was used. An example of measurement is shown in Figure 5a. A histogram was created from the measurement results, which proves that the average size of the internal pore of the powder particles ranges on the border of 15 µm, see Figure 5b. For non-spherical pores, the size of the longest axis was decisive. Pores smaller than 5 µm were manually removed from the measurements, with the mean value and standard deviations then being calculated.

To measure the distribution of grain size, an optical method was chosen using the KEYENCE VHX-5000 microscope. Measurements were carried out at five random locations with an optical magnification of 200x. Each area assessed contained approximately 400 particles from which a static analysis was carried out. Using the statistical software Minitab 17, the cumulative (sum) curve was created, which indicates the number of particles larger or smaller than a certain particle size x. From them were obtained the quantiles d10, d50, and d90, which express the limit to which the size falls to 10, 50, and 90% in the measured particles, see Figure 6 and Table 3.

### 3.2. Evaluation of Curling Angle

The samples were measured on an Alicona optical instrument and the Minitab 17 analytical software was used to evaluate the results. Experimental matrix and results are listed and sorted with parameter settings in Table 4. For each test part, the curling angle was quickly measured ten times. From the measured data the mean value and standard deviation was calculated. The table is also supplemented by a curling angle simulation of four selected samples representing individual representatives in the group. The measured values correspond to the reference value measured in the study [15], which is α = 1.333 ± 0.024°.

### 3.3. Analysis of Taguchi Design

Minitab 17 was used to establish a response table according to Taguchi analysis, where Delta is the difference between the highest and lowest average response values for each factor, and Rank indicates the relative effect of each factor on the response, where No. 1 has the largest effect. A response table (see Table 5) was created for every scanning strategy to find the most significant parameter.

## 4. Discussion

### 4.1. The Influence of Laser Power on Curling Angle

From the response in Table 4, it is apparent that the greatest influence on the formation of the curling angle is the size of the laser power (Rank 1), for all tested strategies. Its influence has a great effect on the development of internal stress in parts and also affects the density of the part; density measurement, however, has not been a subject of this research. Density influence research was conducted in another study [19]. From Figure 7, it is evident that the smallest curling angle was achieved at a value of 300 W and using the CH strategy. On the other hand, the worst results were achieved for a value of 100 W with the M strategy. It is therefore possible to conclude from the study that the lower power has a negative influence on the development of curling angle and hence on the internal stress itself. This is probably due to incomplete melting at low values. Increasing laser power causes the lowering cooling rate, which corresponds with slower laser movement, shorter waiting time and higher preheating. Therefore, the curling angle, residual stresses and thermal gradients are lower [20,21].

### 4.2. The Influence of Field Size on Curling Angle

The length of the individual scanning vectors affects the internal stresses generated by the induction of heat and subsequent quick cooling in the melting pool. According to Kruth et al. [22], a dependency is where a smaller scanning vector delivers a smaller RS. This claim was not confirmed in this experiment, and the best values were obtained at field size of 5 mm^2^; as with Meander, the best hatch distance values were at 5 (0.111 mm). It is therefore advisable to select this size and not to rely on the rule, the smaller the scan vector path is, the less stress applied. Figure 8 shows the dependence of field size of each strategy on the development of curling angle.

## 5. Conclusions

The study examined the effects of laser power, field size and preheating substrate in three commonly available scanning strategies depending on the resulting RS using the BCM method. These were the chessboard, stripes and meander strategies. The test material was made of AISI316L stainless steel powder. The whole experiment was designed and controlled by the DoE according to the Taguchi method. The main findings are:The internal porosity of the powder particles was measured at 99.695%Laser power has the biggest influence on RS regardless of the type of scanning strategy. The ideal setting is a 300 W laser power, which effectively reduces the curling angle.To reduce RS it is ideal to set the field size of the scan vectors to 5 mm^2^ or hatch distance to 0.111 mm.For AISI316L steel, the least influence on RS is a substrate preheating, which in the DoE shows the least significant value.The simulations performed on the selected samples show the same dependence as the samples actually printed, so it is possible to work with them further.

Further research will be testing individual strategies on the resulting mechanical properties and density of the part.

## Figures and Tables

**Figure 1 materials-13-01659-f001:**
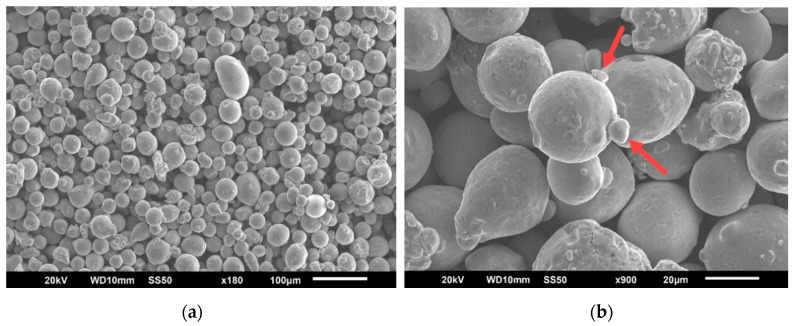
Scanning electron microscopy (SEM) image of 316L metallic powder, magnification ×180 (**a**); satellites with red arrows, magnification ×900 (**b**).

**Figure 2 materials-13-01659-f002:**
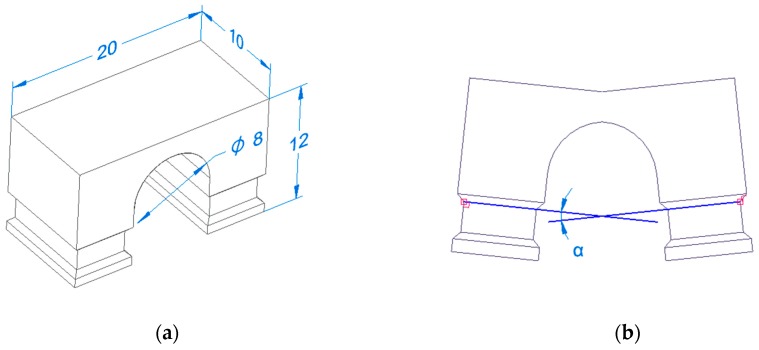
Improved shape of bridge curvature method (BCM) method (**a**); measurement methodology of curling angle α (**b**).

**Figure 3 materials-13-01659-f003:**
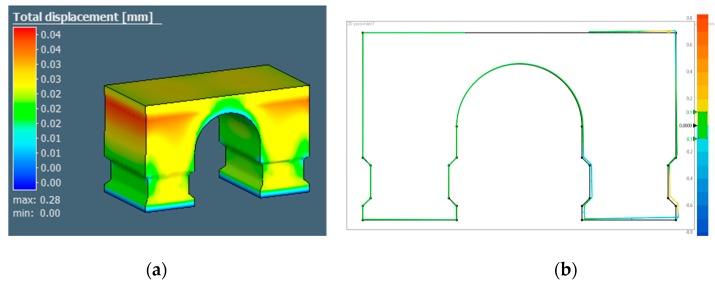
Total displacement done by simulation (**a**); comparison of simulated model and original model (**b**).

**Figure 4 materials-13-01659-f004:**
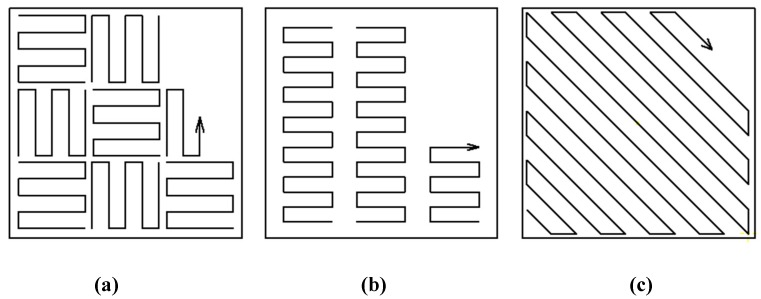
Chessboard strategy (**a**); Stripes strategy (**b**); Meander strategy (**c**).

**Figure 5 materials-13-01659-f005:**
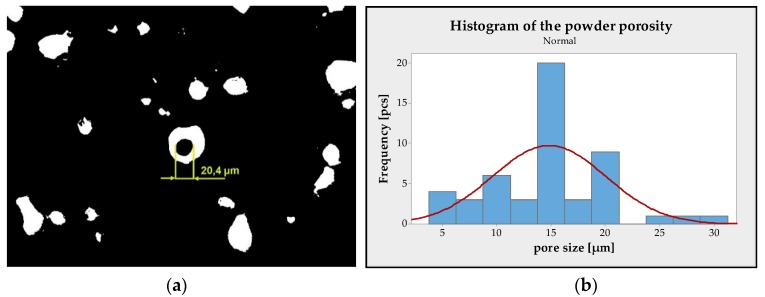
Example of manual measurement of the inner pore (**a**) [18]; histogram of inner pore size distribution in the metallic particle of 316L (**b**).

**Figure 6 materials-13-01659-f006:**
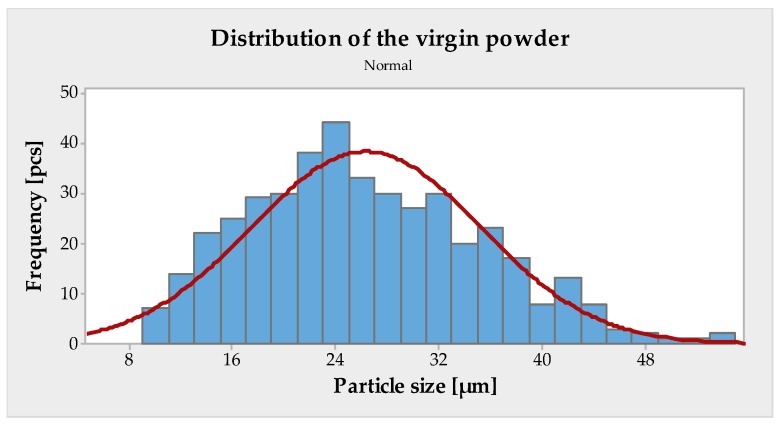
Grain size distribution of 316L material.

**Figure 7 materials-13-01659-f007:**
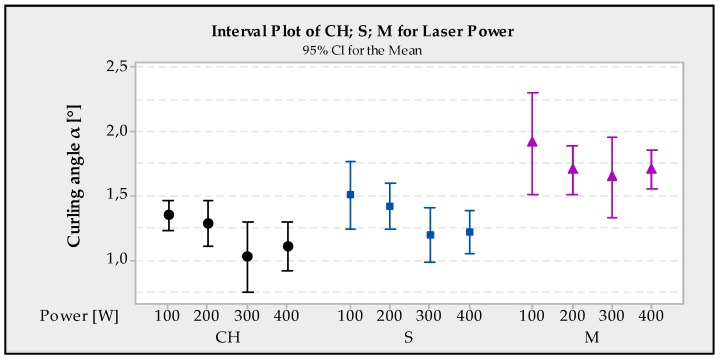
Interval plot of CH, S, M for laser power.

**Figure 8 materials-13-01659-f008:**
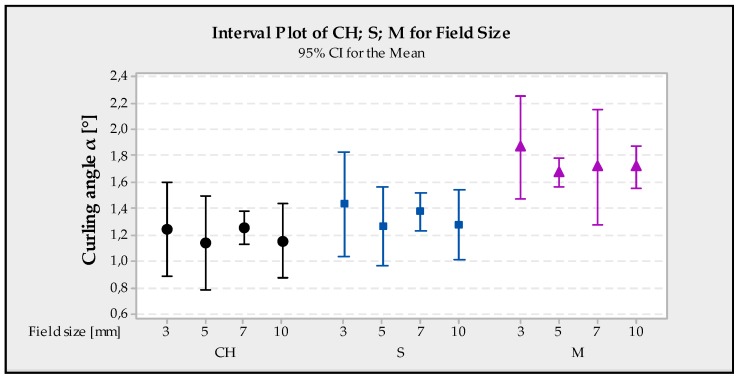
Interval plot of CH, S, M for field size.

**Table 1 materials-13-01659-t001:** Chemical composition of the virgin powder of 316L. All elements are in [wt %].

Fe	Cr	Ni	Mo	Mn	Si	N	O	P	C	S
Balance	18	14	3	< 2	< 1	< 0.1	< 0.1	< 0.045	< 0.03	< 0.03

**Table 2 materials-13-01659-t002:** Variated parameters for DoE.

Parameters	Range of Variables			
**P [W]**	100	200	300	400
**FS/HD [mm]**	3	5	7	10
**T [°C]**	0	170	0	170

**Table 3 materials-13-01659-t003:** Calculated statistical values.

Statistics [μm]	
Mean.	26.33
St. Dev.	8.902
d_10_	19.37
d_50_	25.09
d_90_	32.53
Span	0.525

**Table 4 materials-13-01659-t004:** Design of experiment (DoE) matrix and values of curling angle α for each strategy and simulation.

Sample No.	P [W]	T [°C]	FS/HD [mm]	α CH [°]	α S [°]	α M [°]	α Simulation[°]
1	100	0	3	1.443 ± 0.011	1.744 ± 0.024	2.198 ± 0.036	-
2	100	0	5	1.293 ± 0.028	1.363 ± 0.037	1.703 ± 0.040	1.568
3	100	170	7	1.295 ± 0.015	1.444 ± 0.031	2.051 ± 0.019	-
4	100	170	10	1.382 ± 0.029	1.479 ± 0.042	1.703 ± 0.023	-
5	200	0	3	1.375 ± 0.025	1.511 ± 0.023	1.603 ± 0.046	-
6	200	0	5	1.316 ± 0.036	1.448 ± 0.024	1.587 ± 0.038	1.215
7	200	170	7	1.342 ± 0.041	1.465 ± 0.041	1.801 ± 0.017	-
8	200	170	10	1.121 ± 0.027	1.258 ± 0.049	1.805 ± 0.038	-
9	300	170	3	0.931 ± 0.038	1.183 ± 0.038	1.859 ± 0.036	-
10	300	170	5	0.835 ± 0.032	1.013 ± 0.012	1.744 ± 0.042	1.180
11	300	0	7	1.182 ± 0.031	1.293 ± 0.047	1.408 ± 0.034	-
12	300	0	10	1.157 ± 0.038	1.296 ± 0.035	1.571 ± 0.047	-
13	400	170	3	1.215 ± 0.046	1.292 ± 0.048	1.806 ± 0.023	-
14	400	170	5	1.088 ± 0.015	1.224 ± 0.030	1.648 ± 0.063	1.123
15	400	0	7	1.182 ± 0.040	1.298 ± 0.046	1.608 ± 0.042	-
16	400	0	10	0.948 ± 0.051	1.068 ± 0.055	1.767 ± 0.034	-

**Table 5 materials-13-01659-t005:** Response table according Taguchi Design.

Level	P	FS/HD	T
Chessboard Strategy			
1	1.353	1.241	1.237
2	1.289	1.133	1.151
3	1.026	1.250	-
4	1.108	1.152	-
Delta	0.327	0.117	0.086
Rank	**1**	**2**	**3**
Stripe Strategy			
1	1.508	1.432	1.378
2	1.420	1.262	1.295
3	1.196	1.375	-
4	1.220	1.275	-
Delta	0.311	0.170	0.083
Rank	**1**	**2**	**3**
Meander Strategy			
1	1.914	1.867	1.681
2	1.699	1.670	1.802
3	1.645	1.717	-
4	1.707	1.712	-
Delta	0.268	0.196	0.122
Rank	**1**	**2**	**3**

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
