# Peer review of "Influence of Scanning Strategy Parameters on Residual Stress in the SLM Process According to the Bridge Curvature Method for AISI 316L Stainless Steel"

_materials, 2020, doi:10.3390/ma13071659_

Round 1
Reviewer 1 Report
This research was aimed to study the influence of scanning strategies parameters on residual stress in SLM process according to bridge curvature method for stainless steel AISI 316L. The research subject is interesting and current.
The following aspects should be considered by the authors before the paper can be accepted for publication:
1) The quality of the English is not bad. However, there are some errors along the text. The text should be revised.
2) The objective of the work must be better explained in the Introduction.
3) Although the subject is current, only few references used in the Introduction are from the last 5 years. The authors should improve the description of the state-of-the-art with more recent works.
4) The scales are missing in the micrographs of Fig. 1.
5) The authors identified the studied conditions by numbers (1, 2, 3… 16). Each condition should be clearly presented in the Experimental Procedure (variated parameters + strategy).
Author Response
1) The quality of the English is not bad. However, there are some errors along the text. The text should be revised.
Response 1): The text was revised by native speaker
2) The objective of the work must be better explained in the Introduction.
Response 2): 2 senteces was added to the end part of Introduction to clearly defined the objective of work (see „Track Changes“ in Microsoft Word)
3) Although the subject is current, only few references used in the Introduction are from the last 5 years. The authors should improve the description of the state-of-the-art with more recent works.
Response 3): I tried to improved references as I could, but some of them are crucial for this paper
4) The scales are missing in the micrographs of Fig. 1.
Response 4): The scales were added
5) The authors identified the studied conditions by numbers (1, 2, 3… 16). Each condition should be clearly presented in the Experimental Procedure (variated parameters + strategy).
Response 5): These conditions are given in Table 3, for each strategy were used the same conditions

Reviewer 2 Report
The paper deals with the evermore increasing field of additive manufacturing and aims to evaluate influences of selective laser melting parameters to induced residual stress of the samples printed from AlSi-powder.
In particular, the influence of the direction of laser scanning is investigated, which is likely to have an effect on the printed samples as it influences the alignment and structure of the generated melted joints. Thus, amongst other parameters like speed, temperature and temperature gradients, which influence the size of the molten joints between AlSi-particles, the scanning direction relative to the stress axis may have an important impact. The authors aim to find an optimum combination of laser melting parameters with scanning strategy.
The investigation is made using bridge samples, which exhibit a main axis of deformation if exposed to stress. The deformation could be easily measured via the curling angle between the bridge sockets that is proportional to the residual stress of the sample.
Whilst the topic is of certain interest to the readers and the results are plausible, I would recommend some changes to the manuscript as follows:
The language should be checked and improved (especially grammar).
It would be interesting to add more information about the sample production for people that are not familiar with the method. The paper deals with AM but in section 3.2, an EDM fabricated sample is mentioned. Is this for comparison with AM-made samples?
Second, it is of importance in what manner the sample is made by AM. Does it start with the upper part of the bridge or with the bridge sockets? This is relevant because the forces possibly leading to deformation during manufacturing are dependent on this. The observed deformation is hardly intelligible without this information.
Results should be tabled with standard deviations also.
The discussion should incorporate possible causes that may explain the deviation between modelled values, that show smallest RS for 400 W laser power in contrast to the experiment, that shows 300 W with best values.
This result should then be interpreted as to deliver more insight about the actual processes causing the observed phenomenon in order to gain more generalizable insight. Without these, the results seem to be only applicable to the experimental setup used.
Author Response
The language should be checked and improved (especially grammar).
Response 1: The text was revised by native speaker
It would be interesting to add more information about the sample production for people that are not familiar with the method. The paper deals with AM but in section 3.2, an EDM fabricated sample is mentioned. Is this for comparison with AM-made samples?
Response 2: I improved and add more coments to this topic, see the section 2.2. In text is mentioned EDM only in comparasion with band saw. Band saw was used only to cut samples from substrate (build plate), EDM is used by default.
Second, it is of importance in what manner the sample is made by AM. Does it start with the upper part of the bridge or with the bridge sockets? This is relevant because the forces possibly leading to deformation during manufacturing are dependent on this. The observed deformation is hardly intelligible without this information.
Response 3: This is more specified in section 2.2., (see „Track Changes“ in Microsoft Word)
Results should be tabled with standard deviations also.
Response 4: The standart deviations were added to table with results (see Table 3)
The discussion should incorporate possible causes that may explain the deviation between modelled values, that show smallest RS for 400 W laser power in contrast to the experiment, that shows 300 W with best values.
Response 5: Added more insight about deviation between 300W and 100W laser power and extended discussion see section 4.1.
This result should then be interpreted as to deliver more insight about the actual processes causing the observed phenomenon in order to gain more generalizable insight. Without these, the results seem to be only applicable to the experimental setup used.
Response 6: Thank you for noticing this, I added more sentences explaining the observed phenomen, hope now it is, more clear what happening in this process. Also I think it is applicable not only for experimental use.

Reviewer 3 Report
The manuscript present interesting research results in a field under a continuous development, therefore the readers might have great interest in this possible article.
However, I recommend publishing this manuscript only after several improvements:
- English language must be extensively improved, hard to follow phrases can be found through the entire manuscript
- the references must be updated since about 50% of them are published 10 years ago (or more)
- The modification of the initial BCM method should be more clearly presented with comments regarding the possible influence of these changes on the results.
- at chapter 2.2. Settings and sample fabrication, the phrase "A total of two builds were printed for make samples with substrate preheating and without preheating" is unclear, one might understand that only two samples were printed. I hope I understand well that multiple samples were, in fact, analyzed.
- Subtitles in chapter 4.Discussion must be improved (size of laser power, size of field size)
- if possible the discussion should be extended and compared to other authors results
- might have been interesting to see some XRD patterns on the powder and crystallite size. The powder might be nanostructured with influence on the properties and laser power used. These is however, just an idea, which the authors are free to consider or not.
Author Response
- English language must be extensively improved, hard to follow phrases can be found through the entire manuscript
Response 1: The text was revised by native speaker (see „Track Changes“ in Microsoft Word). Thank you for that, there was a lot to correct
- the references must be updated since about 50% of them are published 10 years ago (or more)
Response 2: I tried to improved references as I could, but some of them are crucial for this paper
- The modification of the initial BCM method should be more clearly presented with comments regarding the possible influence of these changes on the results.
Response 3: Added more comments to this topic, see the section 2.2
- at chapter 2.2. Settings and sample fabrication, the phrase "A total of two builds were printed for make samples with substrate preheating and without preheating" is unclear, one might understand that only two samples were printed. I hope I understand well that multiple samples were, in fact, analyzed.
Response 4: Yes, you understand well, thank you for noticing this. I rewrited the sentence to be more clear.
- Subtitles in chapter 4.Discussion must be improved (size of laser power, size of field size)
Response 5: Subtitles changed to better description section
- if possible the discussion should be extended and compared to other authors results
Response 6: Added more insight about deviation between 300W and 100W laser power and extended discussion see section 4.1.
- might have been interesting to see some XRD patterns on the powder and crystallite size. The powder might be nanostructured with influence on the properties and laser power used. These is however, just an idea, which the authors are free to consider or not.
Response 7: Thank you for that idea, certainly it can be interesting, we will consider it and maybe use it in other article
